# Long-Term Effectiveness of Liraglutide for Weight Management and Glycemic Control in Type 2 Diabetes

**DOI:** 10.3390/ijerph17010207

**Published:** 2019-12-27

**Authors:** Maria Mirabelli, Eusebio Chiefari, Patrizia Caroleo, Biagio Arcidiacono, Domenica Maria Corigliano, Stefania Giuliano, Francesco Saverio Brunetti, Sinan Tanyolaç, Daniela Patrizia Foti, Luigi Puccio, Antonio Brunetti

**Affiliations:** 1Department of Health Sciences, University “Magna Græcia” of Catanzaro, 88100 Catanzaro, Italy; maria.mirabelli@unicz.it (M.M.); echiefari@gmail.com (E.C.); arcidiacono@unicz.it (B.A.); domenicacorigliano@gmail.com (D.M.C.); stefania.giuliano75@gmail.com (S.G.); francescosaverio.brunetti@studenti.unicz.it (F.S.B.); foti@unicz.it (D.P.F.); 2Complex Operative Structure Endocrinology-Diabetology, Hospital Pugliese-Ciaccio, 88100 Catanzaro, Italy; patrizia.caroleo@alice.it (P.C.); puccio55@libero.it (L.P.); 3Division of Endocrinology and Metabolism, Department of Internal Medicine, School of Medicine, Biruni University, 34010 Istanbul, Turkey; stanyolac@gmail.com

**Keywords:** gender difference, liraglutide, weight management, type 2 diabetes

## Abstract

*Background*: Liraglutide is the first glucagon-like peptide-1 receptor agonist (GLP-1 RA) based on the human GLP-1 sequence, with potential weight loss benefits, approved for the treatment of type 2 diabetes (T2D) mellitus. Herein, we aimed to assess the 5-year effectiveness of Liraglutide in the management of weight and glycometabolic control in a Southern Italian cohort of overweight/obese T2D patients, who were naïve to GLP-1 RAs. *Patients and Methods*: Forty overweight or obese patients treated with Liraglutide at doses up to 1.8 mg/day, in combination with one or more oral antidiabetic agents, were retrospectively assessed at baseline, during, and after 60 months of continuous therapy. *Results*: After 5 years of Liraglutide treatment, body weight decreased from 92.1 ± 20.5 kg to 87.3 ± 20.0 Kg (*p* < 0.001), with a mean reduction of 5.0 ± 7.0 Kg and a body mass index (BMI) decrement of −2.0 ± 3.1 Kg/m^2^. On Spearman’s univariate analysis, change in body weight was correlated with female gender and baseline BMI. Hemoglobin A1c (HbA1c) decreased from 7.9 ± 0.9% at baseline to 7.0 ± 0.7% at the end of the study period (*p* < 0.001), followed by a significant reduction in fasting plasma glucose. No significant differences emerged in other biochemical parameters, despite a trend toward improvement in lipid profile. Notwithstanding encouraging effects on several markers of cardiovascular disease (CVD), increments in the 5- and 10-year risk for the first atherosclerotic cardiovascular event were documented, as four incident cases of myocardial infarction. *Conclusions*: Prolonging treatment with Liraglutide can lead to durable benefits in relation to weight and glycemic control, with a greater impact on women. These results extend and corroborate previous observations, suggesting that gender per se may modulate the response to Liraglutide. Despite favorable effects on some established CVD risks factors, the long-term role of Liraglutide in primary prevention of CVD in patients with T2D remains controversial.

## 1. Introduction

Overweight and obesity, defined as an excessive fat accumulation with potential impairing effects on human health [1], represent the major risk factor for type 2 diabetes (T2D) [2]. According to body mass index (BMI) metrics, most individuals with T2D, 80–90%, are classified as overweight or obese [1,3], and thus are recommended to experience either a behavioral or a medication-based weight loss program [4]. In these patients, losing as little as 5% of body weight may improve glycemic control and other cardiometabolic markers, thereby reducing the risk of obesity-related comorbidities, such as cardiovascular disease (CVD) [5,6]. This notwithstanding, loss of body weight and maintenance of weight loss are two of the most difficult challenges obese diabetic patients are faced with, in consideration of the fact that major classes of hypoglycemic agents, such as insulin, sulfonylureas, and thiazolidinediones, have been associated with weight gain [7], with the potential to offset the beneficial effects of glycemic control on CVD risk [8]. In this view, glucagon-like peptide-1 receptor agonists (GLP-1 RAs), by providing an additional benefit of weight loss [9], would represent a preferable second-line option for patients with obesity and inadequately controlled diabetes, as adjunct to lifestyle interventions and metformin [10]. GLP-1 RAs improve glucose homeostasis by enhancing the postprandial insulin secretion and suppressing glucagon release in a glucose-dependent manner. Treatment with GLP-1 RAs can replace the loss in production of the native insulin-releasing hormone GLP-1 from the distal gut, which is a typical feature of T2D [11]. Also, analogously to the gut-derived endogenous hormones, GLP-1 RAs may influence satiety, attenuating appetite sensations and, thus, the amount of food consumed, with immediate consequences on energy balance and weight control [12]. 

Liraglutide, the first long-acting GLP-1 RA based on the human GLP-1 sequence, has been well-detailed in the Liraglutide Effect and Action in Diabetes (LEAD) study program of randomized controlled trials, which explored the efficacy and safety of once daily subcutaneous injections of Liraglutide 1.2–1.8 mg as monotherapy, or in combination with other oral antidiabetic agents [13,14,15,16,17,18]. Although primarily focused on glycemic targets, a significant dose-dependent weight loss was observed in the LEAD trials first [15,18], and subsequently confirmed at the higher 3.0 mg dose in the Satiety and Clinical Adiposity: Liraglutide Evidence (SCALE) program [19]. Also, since its marketing authorization in 2010, a large amount of real-world data have been collected in order to prove the effectiveness of Liraglutide on weight and glycemic control, under routine clinical practice conditions. However, most observational studies have been limited by a short duration of follow-up, usually lasting 2 years or less [20]. 

Herein, we aimed to assess the 5-year clinical effectiveness of Liraglutide 1.2 mg or 1.8 mg in the management of weight and glycometabolic control in a Southern Italian cohort of overweight/obese type 2 diabetic patients attending our endocrinology outpatient clinics, who were naïve to GLP-1RAs. Additionally, we focused on the baseline predictors of drug response in daily clinical practice.

## 2. Materials and Methods

### 2.1. Study Participants

In this retrospective study, we analyzed the long-term efficacy of Liraglutide 1.2 mg or 1.8 mg (*Victoza* ®, Novo Nordisk, Bagsværd, Denmark) in a Southern Italy population of subjects affected by T2D. Data were collected from 40 consecutive diabetic patients, classified as overweight or obese according to BMI status categories (25–29.9 Kg/m^2^ and ≥30 Kg/m^2^, respectively) [1], who were treated with Liraglutide for a minimum follow-up of 60 months, either alone or in combination with other antidiabetic drugs, on the basis of the international clinical practice recommendations for managing hyperglycemia in T2D [21]. Participants were recruited from the Operative Units of Endocrinology and Diabetes (AOU “Mater Domini”, and the AO Pugliese-Ciaccio in Catanzaro) during the period November 2012–August 2019. Patients were excluded if they had switched to Liraglutide from another GLP-1 RA. Age, sex, weight, BMI, blood pressure (BP), lipid profile, fasting plasma glucose (FPG), HbA1c, aspartate aminotransferase/alanine aminotransferase (AST/ALT), serum creatinine with estimated glomerular filtration rate (eGFR), duration of diabetes, micro- and macrovascular complications, and any concomitant pharmacological therapy were recorded at baseline for all participants.

### 2.2. Data Collection

Data collection was approved by the ethics committee of *Regione Calabria Sezione Area Centro* (protocol registry n. 26 of 17 January 2019). All of the data were extracted directly by medical researchers from the patients’ electronic health records and saved into a unified database. As the data were analyzed anonymously, there was no need for written informed consent. Study was performed in accordance with the Declaration of Helsinki.

### 2.3. Assessments

All patients underwent periodical clinical and biochemical evaluations to monitor the efficacy of Liraglutide therapy. The variables analyzed to assess efficacy included: Body weight, BMI, HbA1c, FPG, systolic and diastolic BP, and AST and ALT liver enzymes. Any serious medical problems, including coronary heart disease (CHD) and stroke, were recorded on diary cards and the entries were reviewed at each study visit.

### 2.4. Outcome Measures

The primary outcome was to test the long-term clinical effectiveness of Liraglutide for weight management in our patient cohort with T2D. The primary efficacy outcome measure was the change from baseline in body weight after 60 months of Liraglutide treatment, and the proportion of participants with relative weight loss ≥5%. The secondary outcome measures included changes in HbA1c, FPG, BP, lipid profile, eGFR, and proportion of participants achieving HbA1c level <7.0%. Furthermore, in order to assess the risk for fatal and non-fatal CHD, and fatal and non-fatal stroke, the United Kingdom Prospective Diabetes Study (UKPDS) score (based on gender, age, duration of diabetes, smoking, HbA1c value, systolic BP, total cholesterol/HDL cholesterol ratio, and atrial fibrillation) was calculated [22,23]. Finally, we searched for potential baseline predictors of a better response to therapy.

### 2.5. Statistical Analysis

Initially, continuous variables were tested for normality of distribution using the Shapiro–Wilk normality test. Values of continuous variables are expressed as mean ± standard deviation (SD), and values of categorical variables as numbers and percentages. Either the paired Student’s *t*-test or the non-parametric Wilcoxon signed-rank test was used for within-group longitudinal comparisons, respectively, for normal and non-normal variables. The 2-tailed Fisher exact test was used for comparisons of proportions. Spearman’s rank correlation analysis was used to identify any determinant of Liraglutide efficacy. All significant variables were then forced in a generalized linear model and appropriate covariates were added. A significance level of 0.05 was set for all analyses. Data were analyzed with SPSS 20.0 software (SPSS Inc., Chicago, IL, USA).

## 3. Results

### 3.1. Baseline Characteristics of the Study Cohort

A total of 40 Caucasian diabetic subjects completed 5 years of treatment with Liraglutide: 20 with 1.2 mg/day dosage and 20 with 1.8 mg/day dosage. As reported in Table 1, 55% of the participants were women, the mean of age was 57.5 ± 6.6 years, and the mean duration of diabetes was 8.2 ± 5.6 years. According to BMI categories, 14 patients (35%) were classified as overweight, whereas 26 (65%) had obesity. Women were slightly heavier than man, with a mean BMI of 34.8 ± 8.7 Kg/m^2^ vs. 33.1 ± 3.6 Kg/m^2^, respectively.

As shown in Table 2, when initiating Liraglutide treatment, almost all enrolled patients (97.5%) were on metformin therapy, often in combination with other classes of oral antidiabetic drugs, such as sulfonylureas, meglitinides, acarbose, or pioglitazone. Thirty-seven participants (92.5%) had high blood pressure, and 32 of them were using one or more antihypertensive medications. Thirty-one patients (77.5%) were dyslipidemic and 22 of them were treated with hypocholesterolemic drugs (Table 2).

### 3.2. Primary Outcome

After 5 years of treatment with Liraglutide as a second-line therapy for managing hyperglycemia, a significant weight loss was observed in our patient’s cohort. In detail, body weight decreased from 92.1 ± 20.5 Kg to 87.3 ± 20.0 Kg (*p* < 0.001), with a mean reduction of 5.0 ± 7.0 Kg. As shown in Figure 1, the loss of weight observed during the first 6 months of therapy was maintained throughout the 5-year study period. 

Proportions of patients with a relative weight loss ≥5% were significantly different at 60 and 6 months (47.5% vs. 33.3%, respectively) (*p* = 0.262), suggesting a positive trend. Consistently, BMI decreased from 34.0 ± 11.3 to 32.1 ± 5.8 Kg/m^2^ (*p* < 0.001), with a reduction of −2.0 ± 3.1 Kg/m^2^. On Spearman univariate correlation analysis, the change in body weight was correlated only with female gender (ρ = 0.365, *p* = 0.020) and baseline BMI (ρ = 0.380, *p* = 0.016). Liraglutide dosage was not correlated with weight change.

When tested, in a multiple generalized linear model, the association of female gender with ponderal decrement was increased by introducing baseline body weight at baseline as a covariate (Table 3), whereas it was not modified by age, diabetes duration, and changes in HbA1c levels. Also, no covariates modified the association of BMI with weight change.

### 3.3. Secondary Outcomes

As for body weight, a significant reduction in HbA1c was also observed. Specifically, HbA1c decreased from 7.9 ± 0.9% at baseline, to 7.0 ± 0.7% after 5 years of Liraglutide treatment (*p* < 0.001) (Table 3). This decrement appeared at the 6-month visit and did not show significant variation throughout the follow-up period (Figure 2). Consistently, the proportion of patients achieving American Diabetes Association (ADA) target for glycemic control (HbA1c < 7%) increased from 17.5 to 45.0% at 6 months (*p* = 0.009), reaching 50.0% after 5 years (*p* = 0.004), without significant variations over time. The decrement of HbA1c correlated only with Hba1c value at baseline (ρ = 0.663, *p* < 0.001) and, as expected, with FPG value at baseline (ρ = 0.423, *p* = 0.007).

After 5 years of Liraglutide, FPG decreased from 164.8 ± 32.8 to 140.8 ± 26.6 mg/dL (*p* < 0.01) (Table 4). As for HbA1c, the decrement of FPG was observed at the 6-month visit, and it was maintained until the end of the study. Change in FGP was correlated with either FPG at baseline (ρ = 0.554, *p* < 0.001), female gender (ρ = 0.392, *p* < 0.012), or change in body weight (ρ = 0.362, *p* < 0.022). No significant differences emerged in other biochemical parameters, despite a trend of improvement in lipid profiles (total cholesterol from 180.5 ± 33.3 to 163.8 ± 36.8 mg/dL (*p* = 0.739); HDL cholesterol from 46.6 ± 7.1 to 47.1 ± 8.9 mg/dL (*p* = 0.262); triglycerides from 180.5 ± 33.3 to 142.4 ± 58.5 mg/dL (*p* = 0.283); Table 3).

These results were obtained in our patient population without a substantial intensification of the glucose-lowering therapy and other concomitant medications during the 5-year study period. Also, no differences were reported in the proportion of diabetic microvascular complications. Notably, four incident cases of acute myocardial infarction (MI) and one transient ischemic attack (TIA) were recorded, together with a significant increase in the 5- and 10-year risk of both fatal and nonfatal CHD, and fatal and nonfatal stroke (Table 5). 

Spearman univariate correlation analysis indicated that change in body weight, Liraglutide dosage, or other parameters were not correlated with UKPDS score change.

## 4. Discussion

The results of the current study demonstrate that prolonging treatment of obese and overweight T2D patients with Liraglutide up to 1.8 mg/day for 5 years, in combination with one or more oral glucose-lowering agents, is effective at inducing and sustaining weight loss in a real-world clinical setting. Both baseline BMI and female gender were the main determinants of the decrease in body weight, which occurred independently from other clinical or biochemical features or Liraglutide dosage (1.2 or 1.8 mg). These findings are partially in line with those reported in a Chinese population [24], in which patients with higher baseline BMI values showed greater body weight reduction after 24 weeks of Liraglutide treatment, and in a Northern East Italian cohort [25], where weight loss was related with baseline body weight. Also, the results of this study are in agreement with earlier investigations by our group, which showed the impact of female gender on the glycemic effectiveness of Liraglutide [26]. Gender-dependent influences of Liraglutide on weight loss outcomes extend and corroborate the results of exposure–response analyses of the SCALE program [27], suggesting that ~50% of the difference in body weight loss between men and women could be attributed to higher exposure to Liraglutide in women. The sexual dimorphism in pharmacological responses to GLP-1 RAs has been also investigated in several real-life experiences with the short-acting exendin-4 analogue Exenatide [28,29], all revealing that higher proportions of female participants achieved weight loss targets in comparison to males, consistent with our findings. Although the available evidence in this context is still weak, weight loss appears to have a gender-specific profile following intervention with GLP-1 RAs. However, besides pharmacokinetic considerations, gender might affect the outcome and response to antidiabetic therapies due to distinct adherence rates and divergent behaviors toward drugs [30], other than sex-specific patterns of diabetes and related conditions, including obesity [31,32]. Recently, epidemiological studies [33] reported that Italian diabetic women were 50% more likely to have BMI ≥ 30 kg/m^2^ when compared to their male counterpart, which is congruous with the notice of gender differences among our patients in terms of baseline BMI. The slightly higher degree of obesity in diabetic women may contribute to a greater extent of weight loss after treatment with Liraglutide and other GLP-1 RAs. 

Beside long-term maintenance of weight reduction, the 5-year treatment with Liraglutide lowered HbA1c level by almost 1 percentage point with respect to the baseline value, so that, at the end of the study period, half of participants achieved the ADA goals for glycemic control. The glucose-lowering effect of Liraglutide was independent of the decrement in body weight, albeit greater HbA1c reductions were associated with higher HbA1c baseline values. These results closely resemble those of a recent multicenter observational study [34], reporting the long-term real-world effectiveness of Liraglutide on glycometabolic parameters. In contrast to this latter study [34], however, we did not find any significant change in the cholesterol concentration, notwithstanding a favorable and constant tendency of lipid profile improvement. Differences in background hypocholesterolemic regimens and the small sample size of our patient cohort may explain these discrepancies. Also, we cannot exclude the contribution of population-specific genetic determinants, which may affect distinct blood lipid traits [35] and the response to environmental [36,37] or pharmacological lipid-lowering interventions [38]. 

Additionally, despite significant improvements in glycemic control and obesity indices, together with encouraging effects on lipid profile and systolic BP, considerable increments in the 5- and 10-year risk of fatal and nonfatal CHD, and fatal and nonfatal stroke, were documented after 5 years of Liraglutide treatment in our population, in opposite to previous findings [34]. However, our results in this context are plausible, given that age, a main parameter of the UKPDS score, is the strongest non-modifiable risk factor for CHD and stroke. Also, the impact of aging on cardiovascular risk is more severe in the presence of diabetes, with a transition from a low to a moderate 10-year CHD risk category at only 35 years of age in men and 45 years in women [39]. Thus, age represents a major determinant of the prognostic information included in all estimation risk models [36], whereas modifiable factors, such as body weight, play only a marginal role [22,40]. These considerations may explain our findings of raised CVD risk estimates after 5 years of treatment with Liraglutide, supporting a recent call for novel age- and sex-specific risk prediction models for the assessment of CVD [41]. 

A strength of the present study is the enrollment of overweight/obese diabetic patients who were naïve to GLP-1RAs, as the switch from other molecules (e.g., Exenatide) could have influenced the response to Liraglutide in terms of body weight and HbA1c [42]. Participants were recruited only from endocrinological outpatient clinics within the Catanzaro area, assuring simplified communication channels between practitioners and medical researchers, and small variability regarding data collection. Furthermore, we took advantage of the comparatively genetic homogeneity of the Calabrian population [43,44,45], which may allow for a small inter-individual variability in pharmacological responses [46,47]. Nonetheless, our results can only be applied to patients who show a positive response to Liraglutide and no safety concerns, which would otherwise prevent a 5-year prolonged therapy. Finally, the degree of benefit from Liraglutide in this study should be evaluated with caution, owing to its retrospective observational design and the absence of an active drug comparator as control. 

## 5. Conclusions

In our overweight/obese T2D patients, the 5-year treatment with Liraglutide led to durable benefits in terms of glycometabolic control and weight management, with a greater impact on women than on men. These results extend and corroborate previous observations suggesting that gender may modulate responses to Liraglutide. Despite apparent favorable effects on some established cardiovascular risks factors, such as hyperglycemia and hyperlipidemia, the long-term role of this GLP-1 analogue in primary prevention of CVD in individuals with T2D remains controversial, due to limited available data and potential issues of current methods of CVD risk assessment.

## Figures and Tables

**Figure 1 ijerph-17-00207-f001:**
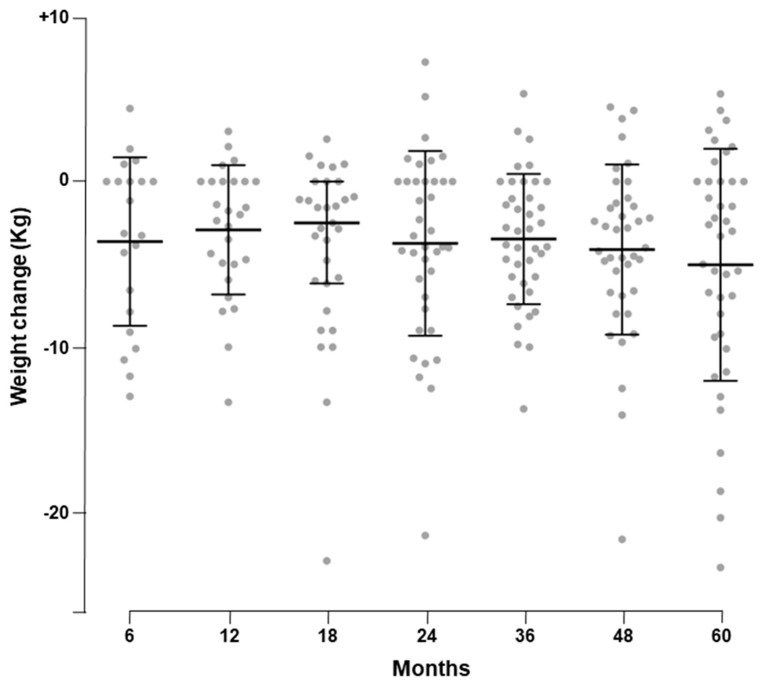
Weight change over the study period.

**Figure 2 ijerph-17-00207-f002:**
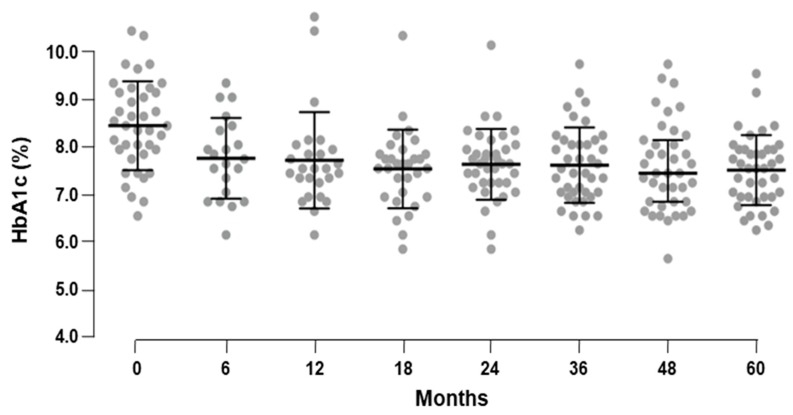
HbA1c change over the study period.

**Table 1 ijerph-17-00207-t001:** Baseline clinical characteristics of the study population (*n* = 40) treated with Liraglutide.

Baseline Characteristics	
Female gender, *n*	22 (55.0)
Ethnicity	Caucasian
Age, years	57.5 ± 6.6
Diabetes duration, years	8.2 ± 5.6
Diabetes duration ≥10 years, *n*	13 (32.5)
Hypertension, *n*	37 (92.5)
Dyslipidemia, *n*	31 (77.5)
Weight, Kg	92.1 ± 20.6
BMI (Kg/m^2^)	34.0 ± 6.8
Overweight (BMI ≥ 25 Kg/m^2^), *n*	14 (35.0)
Obesity (BMI ≥ 30 Kg/m^2^), *n*	26 (65.0)
Coronary artery disease, *n*	5 (12.5)
History of stroke/TIA, *n*	1 (2.5)
Peripheral artery disease, *n*	0 (0.0)
Diabetic microvascular complications, *n*	12 (30.0)
Diabetic retinopathy, *n*	6 (15.0)
Diabetic nephropathy, *n*	5 (12.5)
Diabetic neuropathy (autonomic/peripheral), *n*	1 (2.5)

Data are mean ± standard deviation (SD) or *n* (%). BMI: Body mass index; TIA: Transient ischemic attack.

**Table 2 ijerph-17-00207-t002:** Baseline concomitant medications.

Concomitant Medications	*n* (%)
Metformin	39 (97.5)
Sulphonylureas	6 (15.0)
Meglitinides	3 (7.5)
Pioglitazone	3 (7.5)
Acarbose	2 (5.0)
Insulin	0 (0.0)
Angiotensin-converting-enzyme inhibitors	18 (46.2)
Angiotensin II receptor blockers	14 (35.9)
Calcium channel blockers	9 (23.1)
Beta–blockers	11 (28.2)
Diuretics	17 (43.6)
Loop diuretics	1 (2.6)
Alpha-1-blockers	3 (7.7)
Statins	21 (53.8)
Ezetimibe	5 (12.8)
Cardioaspirin	6 (15.4)

**Table 3 ijerph-17-00207-t003:** Multiple linear regressions for predicting weight decrement.

	B	Beta	T	*p*-Value
BMI	0.387	0.380	2.533	0.016
Female gender	5.086	0.365	2.420	0.020
* Female gender	6.459	0.464	2.975	0.005

* Baseline weight was added as a covariate.

**Table 4 ijerph-17-00207-t004:** Changes in secondary outcomes after 5 years of Liraglutide treatment.

Parameters	Baseline	5 Years	Change	*p*-Value
HbA1c, %	7.9 ± 0.9	7.0 ± 0.7	−0.9 ± 0.9	<0.001
HbA1c < 7%, *n*	7 (17.5)	20 (50)	13	<0.001
FPG, mg/dL)	164.8 ± 32.8	140.8 ± 26.6	−22.7 ± 33.0	<0.001
Creatinine, mg/dL	0.8 ± 0.1	0.8 0.2	−0.1 ± 0.2	0.174 *
eGFR, ml/min/m^2^	92.5 ± 20.3	87.5 ± 17.0	−9.2 ± 17.3	0.787
Total cholesterol, mg/dL	180.5 ± 33.3	163.8 ± 36.8	−13.9 ± 47.5	0.739 *
HDL-C, mg/dL	46.6 ± 7.1	47.1 ± 8.9	1.6 ± 5.8	0.262
Triglycerides, mg/dL	178.3 ± 74.8	142.4 ± 58.5	−24.1 ± 87.0	0.283 *
Systolic BP, mmHg	132.9 ± 15.9	127.5 ± 18.9	−4.6 ± 14.4	0.128 *
Diastolic BP, mmHg	72.9 ± 9.2	72.2 ± 10.1	0.4 ± 11.4	0.983 *

Data are mean ± SD or *n* (%). Either the paired Student’s *t*-test or * the Wilcoxon signed-rank test was employed for continuous values comparisons. Fisher’s exact test was used for comparison of categorical traits. HbA1c: Hemoglobin A1c; FPG: Fasting plasma glucose; eGFR: Estimated glomerular filtration rate; HDL-C: High-density lipoprotein cholesterol; BP: Blood pressure.

**Table 5 ijerph-17-00207-t005:** Changes in UKPDS scores after 5 years of Liraglutide treatment.

UKPDS Score	Baseline	5 Years	*p*-Value
Non-fatal CHD (5 years)	5.0 ± 3.0	6.9 ± 5.8	<0.001
Fatal CHD (5 years)	2.9 ± 2.3	4.6 ± 4.5	<0.001
Non-fatal stroke (5 years)	1.9 ± 1.6	3.8 ± 3.6	<0.001
Fatal stroke (5 years)	0.3 ± 0.2	0.5 ± 0.5	<0.001
Non-fatal CHD (10 years)	11.7 ± 6.5	14.2 ± 9.2	0.004
Fatal CHD (10 years)	7.4 ± 5.3	10.1 ± 8.2	<0.001
Non-fatal stroke (10 years)	5.4 ± 4.5	9.8 ± 7.9	<0.001
Fatal stroke (10 years)	0.8 ± 0.7	1.3 ± 1.1	<0.001
Non-fatal CHD (5 years)	5.0 ± 3.0	6.9 ± 5.8	<0.001
Fatal CHD (5 years)	2.9 ± 2.3	4.6 ± 4.5	<0.001

*p*-values refer to overall differences as derived from non-parametric Wilcoxon signed-rank test. UKPDS: United Kingdom Prospective Diabetes Study; CHD: Coronary heart disease.

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
