# Peer review of "Long-Term Effectiveness of Liraglutide for Weight Management and Glycemic Control in Type 2 Diabetes"

_ijerph, 2019, doi:10.3390/ijerph17010207_

Round 1
Reviewer 1 Report
In the paper entitled: “long-term effectiveness of liraglutide for weight management in tyoe 2 diabetes” Mirabelli and co workers focused their attention on GLP-1 receptor agonist Liraglutide positive effects on body weight and glycaemic control on a cohort of T2D patients enrolled in south of Italy. Five years follow up have been screened by the Authors revealing amelioration due to Liraglutide treatment, overall on T2D female patients.
Paper is well written, rationale is clear and figures are comprehensible. Nevertheless the novelty of these findings is not recognizable on this paper, since a formerly published work by Frison and co workers in 2018 (Clinical Impact of 5 Years of Liraglutide Treatmenton Cardiovascular Risk Factors in Patients with Type 2Diabetes Mellitus in a Real-Life Setting in Italy:An Observational Study already demonstrate the effectiveness Diabetes Ther (2018) 9:2201–2208https://doi.org/10.1007/s13300-018-0503-4) already demonstrated the benefits of 5 years treatment of Liraglutide.
A little novelty can be assessed considering the greater impact on women of Liraglutide therapy, however the small number of patients require an increase in enrolling to better stratify data.
Moreover, Authors affirm that South of Italy has genetic homogeneity, but reference they reported in supporting this concept are slightly inadequate. Authors should provide more detailed references to support their thesis.
Author Response
In the paper entitled: “long-term effectiveness of liraglutide for weight management in tyoe 2 diabetes” Mirabelli and co workers focused their attention on GLP-1 receptor agonist Liraglutide positive effects on body weight and glycaemic control on a cohort of T2D patients enrolled in south of Italy. Five years follow up have been screened by the Authors revealing amelioration due to Liraglutide treatment, overall on T2D female patients.
Paper is well written, rationale is clear and figures are comprehensible. Nevertheless the novelty of these findings is not recognizable on this paper, since a formerly published work by Frison and co workers in 2018 (Clinical Impact of 5 Years of Liraglutide Treatmenton Cardiovascular Risk Factors in Patients with Type 2Diabetes Mellitus in a Real-Life Setting in Italy:An Observational Study already demonstrate the effectiveness Diabetes Ther (2018) 9:2201–2208https://doi.org/10.1007/s13300-018-0503-4) already demonstrated the benefits of 5 years treatment of Liraglutide.
Authors: In our manuscript, we often refer to the article by Frison et al., highligthing the discrepancies between their results and our findings. In details, in lines 230-251, we explored the similarities and differences between these two patients’ cohort, in terms of glycemic outcomes, variations in blood lipids and cardiovascular risk estimates. In addition, our eligibility criteria were more stringent than the criteria used by Frison et al. since patients switching to Liraglutide from others GLP-1 receptor agonists were excluded.
A little novelty can be assessed considering the greater impact on women of Liraglutide therapy, however the small number of patients require an increase in enrolling to better stratify data.
Authors: A recent investigation, exploring diabetes treatment patterns across European countries (Heintjes EM, doi: 10.1016/j.clinthera.2017.09.016), evidenced a very limited use of GLP-1RAs in Italy (~3% as an adjunctive third-line therapy), which is consistent with the last annual reports of the “ARNO Diabetes”, a large multiregional observatory sponsored by the Italian Diabetes Society. Formulary restrictions and practitioners’ preferences refrain most diabetic patients from accessing to novel antidiabetic medications, limiting the sample size of many investigations in this context. In our study, sample size was further limited by the specific design aimed at enrolling only patients from the Catanzaro area and naïve to GLP-1 receptor agonists. These stringent criteria can penalize the sample size but increase the reliability of our results.
Moreover, Authors affirm that South of Italy has genetic homogeneity, but reference they reported in supporting this concept are slightly inadequate. Authors should provide more detailed references to support their thesis.
Authors: According to the reviewer’s request, we now refer to the study by Di Gaetano et al. (An overview of the genetic structure within the Italian population from genome-wide data. Plos One 2012) (Ref. 43).
Reviewer 2 Report
The authors did a good job in presenting the research. One minor edit is suggested: Remove the full stop (.) from the title statement.
Long-Term Effectiveness of Liraglutide for Weight
Management in Type 2 Diabetes.—Remove the full stop (.)
Author Response
The authors did a good job in presenting the research. One minor edit is suggested: Remove the full stop (.) from the title statement.
Long-Term Effectiveness of Liraglutide for Weight
Management in Type 2 Diabetes.—Remove the full stop (.)
Authors: We thank the reviewer for his nice comment. The request has been accomplished.
Reviewer 3 Report
Reviewed manuscript entitled "Long-term effectiveness of Liraglutide for weight management in type 2 diabetes" by Mirabelli et al. is interesting and important. Unfortunately, before acceptance for publication, it needs several revisions.
Minor revisions:
1. Lines 126 and 127. There are "37" and "31". It should be "Thirty-seven" and "Thirty-one", respectively.
2. Title is inadequate to contents of manuscript. There were investigated different parameters, such as BMI, HbA1c, and so on, not only weight. It should be changed.
3. It will be nice, if Authors describe more details on associations between GLP-1 and type 2 diabetes.
Major revisions:
1. Presented results are unclear. Are these results obtained for patients treated only with Liraglutide (alone), or with Liraglutide and other antidiabetic drugs, or are these results as mean for both groups.
2. The aim of study was investigation of long-term effectiveness of Liraglutide for selected parameters important in diabetes. Therefore, the study needs control group (for example patents treated only with Liraglutide, or only with antidiabetic drugs). Lack of these results causes that beneficial? role of Liraglutide remains controversial and unknown.
3. It should be discussed, why Liraglutide has a greater impact on women than on men. Only one phrase on this problem is included in Discussion.
4. It should be also clear presented, how used doses of agonist influence on investigated parameters. Are differences in dependence on doses or not?
5. Discussion should be much richer, because results are interesting.
Author Response
Reviewed manuscript entitled "Long-term effectiveness of Liraglutide for weight management in type 2 diabetes" by Mirabelli et al. is interesting and important. Unfortunately, before acceptance for publication, it needs several revisions.
Minor revisions:
Lines 126 and 127. There are "37" and "31". It should be "Thirty-seven" and "Thirty-one", respectively.Authors: Thank you for the careful review. We have now converted numbers into words.
Title is inadequate to contents of manuscript. There were investigated different parameters, such as BMI, HbA1c, and so on, not only weight. It should be changed.Authors: The title has been slightly modified, as suggested by the reviewer.
It will be nice, if Authors describe more details on associations between GLP-1 and type 2 diabetes.Authors: We have now described these aspects in lines 56-62 of the Introduction section.
Major revisions:
Presented results are unclear. Are these results obtained for patients treated only with Liraglutide (alone), or with Liraglutide and other antidiabetic drugs, or are these results as mean for both groups.Authors: As reported in lines 135-137, 143-144 and 200-205, in this study, we have examined the long-term effectiveness of Liraglutide as a second-line therapy for managing hyperglycemia in type 2 diabetes, which corresponds to International clinical practice recommendations. Given these conditions, when initiating treatment with Liraglutide, the study participants were concomitantly receiving other oral antidiabetic drugs, such as metformin and sulfonylureas.
The aim of study was investigation of long-term effectiveness of Liraglutide for selected parameters important in diabetes. Therefore, the study needs control group (for example patents treated only with Liraglutide, or only with antidiabetic drugs). Lack of these results causes that beneficial? role of Liraglutide remains controversial and unknown.Authors: As it is focused on assessing the effectiveness of liraglutide over 5 years, our study lacks an active comparator group, and the baseline conditions of our patients’ cohort are regarded as the control conditions. This limitation is now discussed in lines 259-263 of the Discussion section.
It should be discussed, why Liraglutide has a greater impact on women than on men. Only one phrase on this problem is included in Discussion.Authors: This aspect is now discussed in more details in lines 213-225 of the Discussion section.
It should be also clear presented, how used doses of agonist influence on investigated parameters. Are differences in dependence on doses or not?Authors: As reported in lines 152-153, 170-171 and 204-205, there are no differences in terms of drug efficacy outcomes between 1.2 and 1.8 mg dosage.
Discussion should be much richer, because results are interesting.Authors: In the light of reviewers’ comments, the discussion section has been extended to better explore the impact of gender on pharmacological responses to GLP-1 receptor agonists.
Round 2
Reviewer 3 Report
Thank you very much for yours responses. They can be accepted.